# Prenatal Counseling throughout Pregnancy: Effects on Physical Activity Level, Perceived Barriers, and Perinatal Health Outcomes: A Quasi-Experimental Study

**DOI:** 10.3390/ijerph17238887

**Published:** 2020-11-29

**Authors:** Shelly Ruart, Stéphane Sinnapah, Olivier Hue, Eustase Janky, Sophie Antoine-Jonville

**Affiliations:** 1Laboratory ACTES EA3596, Univ Antilles, 97159 Pointe-à-Pitre, Guadeloupe, France; stephanesinnapah@gmail.com (S.S.); olivier.hue@univ-antilles.fr (O.H.); sophie.jonville@univ-antilles.fr (S.A.-J.); 2Gynaecology, Obstetrics Department, University Hospital of Guadeloupe, 97159 Pointe-à-Pitre, Guadeloupe, France; eustase.janky@univ-antilles.fr

**Keywords:** physical activity, counseling, barriers to physical activity, prenatal care, outcomes

## Abstract

Physical activity during pregnancy has many health benefits. However, the physical activity level is insufficient throughout pregnancy and women report perceived barriers to physical activity. This study assessed the impact of a counseling intervention offered in addition to routine pregnancy care on physical activity patterns, perceived barriers, and perinatal health outcomes. A quasi-experimental trial was conducted in the Maternity Unit of a hospital in Guadeloupe (a French department). Ninety-six pregnant women were allocated to a control or intervention group. Regular physical activity counseling was dispensed to the women in the intervention group by trained healthcare providers. The physical activity level and the perceived barriers were assessed in each trimester. Outcomes for the perinatal health of the mother and child were measured throughout pregnancy and after delivery. The perceived barriers, such as a lack of information about the health benefits and risks over the two trimesters (all *p* < 0.05) and insecurity related to practice throughout pregnancy (all *p* < 0.05), were different in favor of the intervention group. There were no significant between-group differences for the major indices of physical activity, whether measured or reported. The intervention women reported significantly more sedentary activity compared with the control group in the third trimester, 64.7 (36.4–78.7) vs. 22.7 (9.4–49.8) MET-hours/week, respectively (*p* < 0.001). The perinatal health outcomes for the mother and child showed no significant differences. The intervention was unable to limit the decline in physical activity or improve health outcomes. However, it was associated with an improvement in the perception of barriers. Future research should focus on interventions that have a sufficient quantitative impact on perceived barriers in order to limit physical activity decline.

## 1. Introduction

Research has shown that advice and information provided by health professionals can influence the physical activity (PA) behaviors of pregnant women [1]. Yet, appropriate counseling is still insufficient [2], especially for pregnant women with weight problems. The literature reports PA declines throughout pregnancy [3]. Women report a lack of knowledge and information on the recommendations for PA as barriers to practice [4]. In Guadeloupe, women of childbearing age are overweight and do not meet the recommendations for PA [5,6]. The positive influence of PA during pregnancy on mother–child outcomes has gained increased attention from public health authorities [7], particularly because PA is a modifiable behavior [8].

We proposed a feasible, low-cost counseling intervention as part of the routine pregnancy care, with the aim of increasing low PA levels. The first objective of the study was to evaluate the results of this intervention by comparing the PA patterns of the participants to those of the women receiving routine care in the same unit. The secondary objective was to investigate whether the intervention impacted the perception of barriers and the outcomes for the perinatal health of the mother and child.

## 2. Materials and Methods

### 2.1. Study Design

The study was based on a PA promotion project targeting maternal and child health including individual and lifestyle counseling throughout pregnancy. The PA promotion project was based on several elements reported in the literature, i.e., the low level of PA during pregnancy, and the benefits of regular PA on maternal and fetal health, on the one hand. On the other hand, the project aimed to reduce the perceived barriers declared by pregnant women and the lack of knowledge and dispensation PA recommendations by health professionals. The PA promotion project was based on the hypothesis that reducing barriers and improving levers to PA could reduce the decline in PA levels during pregnancy. A longitudinal quasi-experimental repeated-measures design was employed. The project was presented to the midwives and gynecologists of the Maternity Unit of the Basse-Terre Hospital (Guadeloupe, FWI). Before beginning the research project, health professionals were randomly assigned to the control group or the intervention group. The health professionals who randomized in the intervention group attended two meetings, each lasting approximately 30 min, during which time they received information about the PA recommendations. They were also given a document summarizing the recently updated American College of Obstetricians and Gynecologists (ACOG) recommendations (on PA: recommended and non-recommended activities, signs of cessation of PA practice) [9,10]. A follow-up sheet was created so that the project coordinator and the health professionals in the intervention group could communicate to ensure consistency in the message about the PA recommendations. No recommendations were made to the control group practitioners, who were free to talk about PA with their patients or not.

### 2.2. Participants and Ethics

The inclusion criteria were as follows: maternal age ≥ 18 years old, a single fetus, gestation ≤ 15 weeks, French speaking and possessing a mobile phone (for the communication with the project coordinator). Women with medical or obstetric complications or increased risks were excluded from participation, in line with the ACOG guidelines. All participants provided written informed consent. The study was conducted in accordance with the Declaration of Helsinki and the current local regulatory obligation. The database for this study has been registered under No. MR 5815250919.

### 2.3. Recruitment and Randomization

At the end of each standard antenatal appointment, if a health professional deemed a woman eligible to participate according to the inclusion criteria, the coordinator was invited to explain what the research entailed (information on the project objectives and procedures). Following this explanation, the pregnant women were allocated to either the intervention or control group, depending on the gynecologist who was following them. All health professionals had a preliminary presentation of the project, and they were randomly assigned to the recruitment of control group or intervention group. Women were recruited between January 2017 and April 2018. None changed groups during the follow-up.

### 2.4. Physical Activity Intervention

Each woman in the intervention group monthly received PA consultations with the trained PA teacher. Consultations were held alongside the routine monthly pregnancy visits with the health professionals. In a session, the women were first encouraged to comply with the ACOG recommendations: At least 20–30 min of PA per day on most or all days of the week. Consultations were individually tailored. For example, women who had previously been engaged in no or very little moderate-intensity PA were advised to begin with 15 min of moderate-intensity PA, gradually increasing to 30 min, with three-times per week frequency.

Discussions were then held on the recommended and non-recommended activities, as well as the risks and benefits of PA related to pregnancy. In addition, the barriers usually reported by women to justify their lack of PA practice were discussed. Before or after this PA consultation and within their routine, the gynecologists and midwives were also instructed to systematically devote 1 min of their time to providing information or advice about PA (using the follow-up sheet) in order to supplement and/or reinforce the messages delivered by the coordinator. This time could be lengthened if necessary.

### 2.5. Control Group

The women in the control group received standard antenatal care. Control group participants were asked to complete the same measures as the intervention group at the same time points.

## 3. Data Collection

At each end-of-trimester visit, between 12–15 weeks, 24–28 weeks, and 35–37 weeks, PA and claimed barriers to PA were assessed.

### 3.1. Physical Activity Behavior

PA behavior was measured objectively with an activity monitor (Polar A300 France) and subjectively with the adapted French version of the pregnancy physical activity questionnaire (PPAQ) [11].

After receiving the instructions, the women were asked to wear the activity monitor for 7 consecutive days at each end of trimester from morning till night, with permission to remove it before bedtime. The activity monitor was validated and found to be reliable for adult women [12].

The number of days wearing the monitor and the amount of time wearing it per day varied between the participants and the sample size was not consistent for statistical analysis. We therefore decided it would be more appropriate to decrease these recommendations. Data were thus retained for analysis if the accelerometers were worn at least 8 h per day and for at least 3 days, including 1 day on the weekend [13].

The women reported the duration (per day or week) they spent on each activity (data from PPAQ). Each activity was then assigned an intensity value, based on the values found in the compendium of physical activity [14]. The unit used to characterize the intensity was the metabolic equivalent of task (MET, where 1 MET = energy expended at rest). The time spent in each activity was multiplied by its intensity to obtain an average of the weekly energy expenditure MET-hours/week (MET-h/week), then added to calculate the weekly total activity.

### 3.2. Perceived Barriers to Physical Activity

There is a large literature about the negative impact of pregnancy on behaviors and PA levels in women, however insufficient understanding persisted as to the factors behind this decline in PA during this period. The literature has focused on the factors that influence these PA behaviors, and it emerges that pregnant women perceive a series of barriers to their practice [15]. Previously, studies on predictors of physical activity during pregnancy have mainly focused on demographic and unmodifiable barriers, for example with pregnancy-related nausea, pregnancy-related fatigue or labor [16]. Nevertheless, the literature more recently identified modifiable perceived barriers [4] and several studies have proposed a classification of these barriers, into three categories: environmental, intrapersonal and interpersonal [15,17,18,19]. To our knowledge, there is not a validated questionnaire on perceived barriers to PA in pregnant women, however studies such as that by Haakstad et al. have used a structured questionnaire based on elements of existing literature [20]. A structured questionnaire on the barriers to PA in pregnant women was developed in line with the literature [4]. The 25 questions were classified into main groups (intrapersonal, interpersonal and environment), as observed in the literature. This questionnaire was based on a 5-point Likert scale ranging from 1 (strongly disagree) to 5 (strongly agree).

### 3.3. Outcomes

Sociodemographic characteristics and obstetrical history were collected from the medical records. Secondary results were evaluated, such as the proportion of participating women who showed excessive gestational weight gain (excessive GWG), the evolution of glycemia, the incidence of gestational diabetes during pregnancy, mode of delivery and neonatal outcome.

Maternal weight was measured during the monthly antenatal appointments using an electronic scale (Seca 861 Class III Scale). The total GWG was calculated from the maternal weight. Based on the ACOG recommendations [21], total GWG was defined as excessive if above the upper limit determined for each body mass index (BMI) class: 18.0, 16.0, 11.5 and 9.0 kg in pre-pregnancy underweight, normal weight, overweight and obese mothers, respectively. The weight gain was subsequently considered not excessive if it was within the ACOG recommendations (12.5–18, 11.5–16.0, 7.0–11.5 and 5.0–9.0 kg, respectively). Pre-pregnancy BMI was calculated using pre-pregnancy height and weight and the women were categorized as underweight, normal weight, overweight or obese. The participants underwent two fasting plasma glucose tests between 13 and 15 weeks and 35 and 37 weeks and 75 g oral glucose tolerance testing (OGTT) at 26–28 weeks’ gestation.

For glycemic control and diagnosis of gestational diabetes, we used the diagnostic criteria recommended by the International Association of Diabetes and Pregnancy Study Groups (IADPSG) [22]. Hypertension was measured during the monthly antenatal appointments. Hypertension was defined as diastolic blood pressure (BP) ≥ 90 mm Hg and systolic BP ≥ 140 mm Hg, based on the average of at least two measurements, and recorded in the medical file. The outcome was the number (sample/percentage) of women who developed hypertension during early labor.

Mode of delivery was either vaginal or cesarean. Gestational age at delivery, birth weight (kilograms) and birth length (meters) were collected. Neonatal anthropometric values were collected postpartum from the medical records. The same data were collected at 1 month and 2 months by telephone following the pediatric visit.

## 4. Statistical Analysis

The general characteristics of the two study groups were first described using means and standard deviations (SD) for continuous variables and frequency for categorical variables. They were then compared using independent t-tests for continuous variables and the chi-square or Fisher’s exact test for categorical variables.

Repeated measures analysis of variance (ANOVA) was used to investigate changes between groups over time for PA behaviors and perceived barriers. Mann–Whitney U tests were used when the conditions of application were not met. A post-hoc Bonferroni multiple comparison procedure was then used. For the Likert scales, responses were recoded as “positive,” “negative” or “neither agree nor disagree.” Variables and statistics are presented as means (SD), frequencies (*n*) or medians (quartiles).

For the neonatal outcomes collected monthly from delivery to 2 months, repeated measures ANOVAs were used.

Sample size calculations were based on the efficacy to detect a difference in MET-h/week on PA behavior with randomization according to a 1:1 ratio. We used published values for pregnant women with the PPAQ as a reference, and a standard deviation was chosen: ±70 for the two groups. The analysis of the study size calculation was implemented with G*Power 3.1, with an alpha threshold of 5% and a power of 80% to 90%. The required sample size of 44 participants in each study group was reached.

All analyses were performed using IBM SPSS statistics 23. A *p*-value of <0.05 was considered statistically significant.

## 5. Results

Figure 1 shows the flow of participants through the study. One hundred and seventy-two women were assessed for eligibility for the study and data are presented for 96 or 32 women, depending on the indices. The mean gestation at recruitment was 11.9 ± 2.4 weeks at routine antenatal appointments. An overview of the baseline characteristics is given in Table 1. The two groups were comparable on measured and reported parameters, except marital status and educational level.

### 5.1. Barriers to PA

Table 2 shows the responses to questions on the perceived intrapersonal, interpersonal and environmental barriers to PA in the intervention and control groups. When the responses from the two groups were pooled, agreement significantly increased from the first to the third trimester on two points: both the weight gain related to pregnancy (trimester effect: *p* < 0.001) and appearance changes over the course of pregnancy (*p* = 0.035) can be barriers to PA.

The between-group comparisons showed that barriers were lower in the intervention group, significantly so in one or more trimesters. For example, the feeling of insecurity when practicing PA was lower in the intervention group (i.e., *p* = 0.027, *p* = 0.007, *p* = 0.008, respectively) in the three trimesters, as was weight related to pregnancy in the second trimester (*p* = 0.030).

### 5.2. Self-Reported Physical Activity

The median and quartiles of energy spent in PA by intensity and type are shown in Table 3 and Table 4. For both groups pooled, the self-reported PA significantly decreased from the first to the second, the second to the third, and the first to the third trimesters of pregnancy (Table 3).

The intervention women reported significantly more sedentary activity compared with the control group in the third trimester (*p* < 0.001) (Table 4). There was no significant difference between groups for other intensities and types of activities.

### 5.3. Measured Physical Activity

The activity monitor wear time did not differ between groups or across trimesters (*p* > 0.05). Repeated measures ANOVAs indicated a main effect of trimester for walking for all women: 520 ± 45.9, 440 ± 39.7, 354 ± 23.1 steps/hour respectively, in the first, second and third trimesters. The post-hoc test indicated that the decreases occurred between first and third trimesters (*p* = 0.002) and second and third trimesters (*p* = 0.025). There was no group effect (*p* = 0.2) or group x trimester effect (*p* = 0.7) (Table 5).

### 5.4. Maternal Outcomes

Maternal outcomes are summarized in Table 6. The proportion of women with excessive GWG was 39.8%, with no significant difference between groups (*p* = 0.5). Subgroup analyses according to pre-pregnancy BMI also provided no evidence of differences in excessive GWG between the intervention and control groups.

We also assessed adherence to the ACOG recommendations according to the pre-pregnancy BMI subgroups (overweight vs. non-overweight BMI, kg/m^2^) and found that excessive GWG was more frequent among overweight women compared to normal-weight participants (75.8% vs. 24.2%) (Odds ratio (OR) 0.25, 95% Confidence Interval (CI) (0.95–0.66)), (*p* = 0.004) (Figure 2).

The incidence of gestational diabetes was 15.8% (12/76), with no significant (OR 2.41, 95% CI (0.66–8.83)), (*p* = 0.1 in the intervention group with control as the reference).

There was no time effect (*p* = 0.6) or group x time effect (*p* = 0.3) on fasting plasma glucose throughout pregnancy (trimester 1, trimester 2, trimester 3). A group effect on fasting plasma glucose was noted, with higher values in the intervention group (*p* = 0.04).

We observed no group effect (*p* = 0.5) or group x time effect (*p* = 0.3) on the glucose concentration measured during the OGTT.

No significant differences were observed between the two groups regarding the mode of delivery (cesarean), the area under the curve (AUC) and maternal hypertension during labor (*p* = 0.3, *p* = 0.7, *p* = 0.3, respectively). There was no significant difference in maternal hypertension between groups throughout pregnancy (*p* > 0.3).

### 5.5. Neonatal Outcomes

Neonatal outcomes are summarized in Table 7. There was no significant difference in the gestational age at birth (*p* = 0.7) between the two groups.

There was no group effect (*p* = 0.1) or group x age effect (*p* = 0.1) on weight from birth until the second month. There was no group effect (*p* = 0.1) or group x age effect (*p* = 0.1) on length from birth until the second month. The rate of preterm births was low (5.4%) and did not significantly differ between groups.

## 6. Discussion

This quasi-experimental study investigated the benefits of an intervention project to encourage PA during pregnancy as part of the routine prenatal care. The pregnant women in the intervention group who had received counseling on PA reported fewer barriers to PA than the control group. However, no significant improvement in major PA behaviors or health outcomes was evidenced in this group.

Perceived barriers to PA practice appear to be consistently associated with the PA behaviors of pregnant women [17,23]. The modulation of these barriers was thus considered as an interesting lever to target in our intervention. The intervention and control groups showed differences in their responses to questions about the perceived intra- and inter-personal barriers (Table 2), which can be interpreted as an improvement in response to the intervention. These results agree with a recent large-scale study with fewer intrapersonal (not health-related) barriers to leisure-time PA in the intervention group [20]. Pregnant women regularly report that a major barrier to PA is fear for their personal health and that of their baby; correspondingly, the health of their baby is a major motivating factor [23]. In our study, insecurity related to PA was systematically lower in the intervention group. This might be related to the lower number of reports of insufficient information on both PA benefits and risks in this group and provides putative evidence of program efficacy, as women exposed to a PA education campaign were found to be much more likely to report information on PA [24].

Our intervention appears to have changed the representations of the women since the frequency of the items declared as barriers was lower in the intervention group. However, we cannot exclude the possibility that group differences in educational level and stage of pregnancy at recruitment contributed to the differences in the perceived barriers (Table 1), particularly in the first trimester.

The main program outcome was the women’s PA behavior. This study provides little or no evidence of the program’s effectiveness, whether the indices refer to reported or measured PA.

We observed a general decrease in self-reported PA throughout pregnancy, which is in line with the literature, in particular those studies using the PPAQ [25]. The pattern we reported in the intervention group is quite similar to that observed in other studies [26] and is characterized by involvement in sedentary activities, especially in late pregnancy. Sedentary activities are very low-intensity activities, with no demonstrated impact on health [27]. In our study, a decline was evidenced in the control group even for that intensity category, but it was preserved in the intervention group. This might suggest a positive effect of the intervention, most likely related to the intervention group’s greater awareness of the PA benefits during the late pregnancy visits (Table 2). However, this was not reflected by the reports of involvement in moderate-intensity PA, which was the target of counseling, or by the objective PA data.

Wrist accelerometers were chosen for this study as their superiority for objective measurement of PA during pregnancy has been demonstrated [28]. However, compliance was only 33.3% in this study even though, in accordance with previous research with pregnant women [3], the criterion for valid wear time was reduced to 8 h per day, which is lower than in PA research with non-pregnant groups.

The number of steps per hour declined in the last trimester (Table 4) in both groups, in accordance with the PPAQ data. However, no group or interaction effect was significant, suggesting that the intervention was not effective. Although associations have been systematically/regularly reported between perceived barriers and PA practice [17], counseling aimed at overcoming barriers has not succeeded in improving the PA patterns of pregnant women. Indeed, to our knowledge, all such interventions integrated into routine care have failed to significantly limit the decrease in PA over the course of pregnancy [3,29].

Excessive GWG and other metabolic indices were examined as secondary outcome parameters characterizing the women’s health. Participants with overweight before pregnancy were more likely to develop excessive GWG (Figure 2), in agreement with the literature [21,30]. The intervention and control groups had similar pre-pregnancy weight status distribution. The proportion of excessive GWG was also comparable between groups (40.0% and 39.5% in the intervention and control groups respectively, Table 5). This result might be interpreted as a weakness of our program since association studies have reported lower odds of excessive GWG in women following the PA advice of their obstetric providers [31].

However, another intervention study also failed to significantly reduce excessive GWG (*p* = 0.7), with 45.1% and 45.7% of the women in the intervention and control groups respectively, showing excessive GWG [32]. These results confirm the challenge of improving maternal outcomes through interventions.

A similar pattern with no group difference was found for the indices of glucose metabolism, except fasting plasma glucose, which was significantly higher in the intervention group. The actual cause and meaning of this significant effect are hard to determine, especially since this effect was isolated. For example, the OGTT results were similar and there was no between-group difference for the diagnosis of gestational diabetes, although it should be noted that the study was not powered to measure prevalence.

The mode of delivery (vaginal or cesarean), maternal hypertension and neonatal outcomes were not affected by the intervention. This is unsurprising since in the randomized controlled trials on exercise throughout pregnancy showing positive maternal and neonatal results [33], the women in the intervention groups were generally involved in supervised PA with sufficient intensity (at least moderate), duration (about 50–60 min/session) and frequency (three times per week).

### Limitations and Strengths of the Study

It should be noted that our database was incomplete due to the difficulties inherent to follow-up, particularly during pregnancy. Also, we did not verify the quality or quantity of information provided by the health professionals in the intervention group, which is an important point since discrepancies with ACOG’s guidelines are frequent, especially for exercise intensity [34]. The rather brief time dedicated to training for the health professionals in order to prepare them to counsel the women might also be viewed as a limitation, although it should be recalled that we were looking for alternative solutions to interventions requiring high resources, such as supervised exercise programs. Although a large part of the studies on the PA of women during pregnancy has focused on non-athletic women [35], the objective of increasing or maintaining PA levels cannot be applied to all women, especially elite female athletes who already engage in vigorous PA and for whom a decline in PA during pregnancy is probably necessary [10,36,37].

The strengths of this study include the follow-up throughout pregnancy and the attention to outcomes related to perceived barriers, PA behaviors and maternal and child health indices. Another strength is that the counseling was integrated into the prenatal care program, which we expected would yield a high participation rate [38].

## 7. Conclusions

Although the PA counseling intervention provided as part of routine care was unable to limit the decline in PA throughout pregnancy, it was associated with an improvement in the perceived barriers to PA practice. Research on how to adapt counseling to better address the perceived barriers reported by women, individually and throughout pregnancy, seems required. Such research might serve to enrich the training of health professionals with regard to physical activity benefits.

## Figures and Tables

**Figure 1 ijerph-17-08887-f001:**
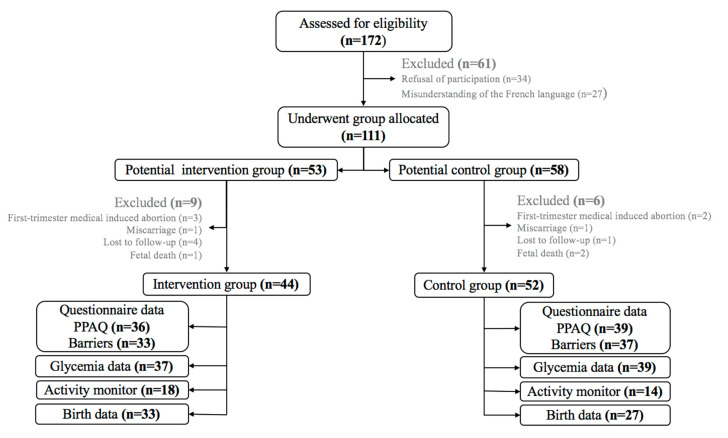
Flowchart of participant progress. Data collected throughout pregnancy.

**Figure 2 ijerph-17-08887-f002:**
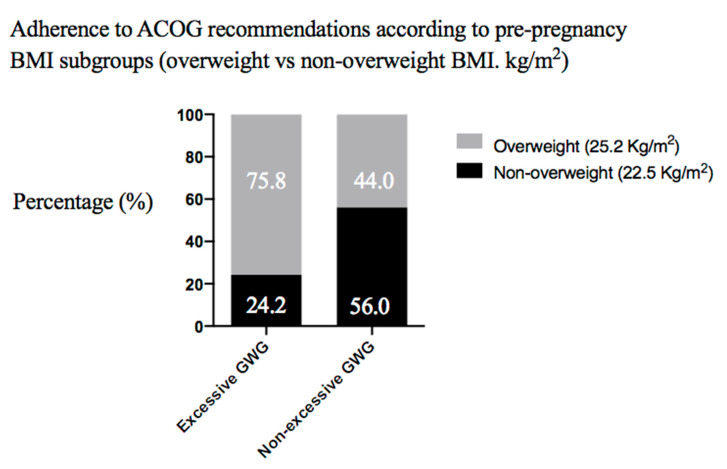
Excessive and non-excessive gestational weight gain (GWG) in women according to body mass index (BMI, kg/m^2^) category (non-overweight vs. overweight).

**Table 1 ijerph-17-08887-t001:** Baseline characteristics of study participants (*n* = 96, unless otherwise stated) and subgroup with additional data on activity monitor (*n* = 32, unless otherwise stated). *p* stands for the intervention vs. control group comparison.

	Study Participants				Subgroup of Participants with Activity Monitoring			
	All, *n* = 96	Intervention (*n* = 44)	Control (*n* = 52)	*p*	All, *n* = 32	Intervention (*n* = 18)	Control (*n* = 14)	*p*
Age, years	29.0 ± 6.7	29.4 ± 6.4	28.7 ± 6.8	0.6	28.3 ± 6.1	28.8 ± 5.7	27.6 ± 6.9	0.2
Height, m	1.64 ± 0.06	1.63 ± 0.06	1.64 ± 0.06	0.4	1.64 ± 0.05	1.64 ± 0.05	1.63 ± 0.06	0.3
Gestational age at entry, weeks	11.9 ± 2.4	11.3 ± 2.4	12.4 ± 2.2	0.02	11.8 ± 2.1	11.2 ± 2.4	12.6 ± 1.5	0.06
Pre-pregnancy weight, kg	72.3 ± 17.4	71.7 ± 20.3	72.8 ± 14.6	0.7	71.8 ± 17.2	71.6 ± 18.0	72.1 ± 16.9	0.4
Pre-pregnancy BMI, kg/m^2^	26.7 ± 5.8	(*n* = 41) 26.6 ± 6.2	(*n* = 45) 26.9 ± 5.5	0.4	25.9 ± 7.5	(*n* = 17) 25.1 ± 8.6	26.9 ± 6.0	0.4
Pre-pregnancy BMI category, *n* (%) -BMI < 18.5 kg/m^2^	6/86 (7.0) [0.0–29.4]	2/41 (4.9) [0.0–11.6]	4/45 (8.9) [0.0–17.3]	0.7	3/31 (9.7) [0.0–20.3]	1/17 (5.9) [0.0–17.4]	2/14 (14.3) [0.0–33.3]	0.7
-BMI 18.5–24.9 kg/m^2^	31/86 (36.0) [18.8–53.2]	17/41 (41.5) [26.2–56.8]	14/45 (31.1) [17.4–44.8]		9/31 (29.0) [12.8–45.2]	6/17 (35.3) [11.9–58.7]	3/14 (21.4) [0.0–43.7]	
-BMI 25.0–29.9 kg/m^2^	22/86 (25.6) [6.9–44.3]	10/41 (24.4) [11.1–37.7]	12/45 (26.7) [13.6–39.8]		9/31 (29.0) [12.8–45.2]	5/17 (29.4) [7.1–51.7]	4/14 (28.6) [4.0–53.2]	
-BMI 30.0–40.0 kg/m^2^	27/86 (31.4) (13.6–49.2)	12/41 (29.3) (15.2–43.4)	15/45 (33.3) [19.4–47.2]		10/31 (32.3) [15.6–49.0]	5/17 (29.4) [7.1–51.7]	5/14 (35.7) [9.7–61.7]	
Marital status, *n* (%) -Single	29/79 (36.7) [26.0–47.4]	9/39 (23.1) [9.7–36.5]	20/40 (50.0) [34.3–65.7]	0.01	9/29 (31.0) [13.9–48.1]	2/17 (11.8) [0.0–27.6]	7/12 (58.3) [29.2–87.4]	0.01
-Living with partner/married	50/79 (63.3) [52.6–74.0]	30/39 (76.9) [63.5–90.3]	20/40 (50.0) [34.3–65.7]		20/29 (69.0) [51.9–86.1]	15/17 (88.2) [72.4–100.0]	5/12 (41.7) [12.6–70.8]	
Educational level, *n* (%) -Higher Education	32/79 (40.5) [29.6–51.4]	20/39 (51.3) [35.4–67.2]	12/40 (30.0) [15.6–44.4]	0.02	15/29 (51.7) [33.2–70.2]	12/17 (70.6) [48.3–92.9]	3/12 (25.0) [0.0–50.6]	0.04
-Secondary	19/79 (24.1) [14.6–33.6]	11/39 (28.2) [13.9–42.5]	8/40 (20.0) [7.4–32.6]		4/29 (13.8) [1.0–26.5]	2/17 (11.8) [0.0–27.6]	2/12 (16.7) [0.0–38.7]	
-Before secondary	28/79 (35.4) [24.8–46.0]	8/39 (28.6) [14.2–43.0]	20/40 (50.0) [34.3–65.7]		10/29 (34.5) [17.6–53.1]	3/17 (17.6) [0.0–36.3]	7/12 (58.3) [29.2–87.4]	

Body mass index: BMI.

**Table 2 ijerph-17-08887-t002:** Perceived barriers to physical activity during pregnancy *n* (%) for all such values.

	Intervention			Control		
	First Trimester	Second Trimester	Third Trimester	First Trimester	Second Trimester	Third Trimester
**Intrapersonal**						
*Negatively related*						
Weight related to pregnancy	1 (3.0)	4 (12.1) *	10 (31.3)	3 (8.1)	12 (32.4)	17 (45.9)
Insecurity related to practice	11 (33.3) *	9 (27.3) *	9 (28.1) *	22 (59.5)	22 (59.5)	21 (56.8)
Fatigue after work	14 (42.4) *	13 (39.4)	16 (50.0)	24 (64.9)	20 (54.1)	22 (59.5)
*Positively related*						
Being motivated	28 (84.8)	27 (81.8) *	26 (81.3) *	25 (67.6)	20 (54.1)	17 (45.9)
**Interpersonal**						
*Negatively related*						
Lack of information about benefits and risks, *n* (%)	7 (21.2)	4 (12.1) *	1 (3.1) **	7 (18.9)	14 (37.8)	12 (32.4)
Lack of friendly support	6 (18.2) *	7 (21.2) *	7 (21.9)	15 (40.5)	17 (45.9)	15 (40.5)
**Environment**						
*Negatively related*						
Lack of sports facilities	6 (18.2) *	7 (21.2) *	15 (46.9)	15 (40.5)	15 (40.5)	17 (45.9)

Data on perceived barriers: *n* = 33 and 37 for the intervention and control groups, respectively; * *p* < 0.05 compared to control group; ** *p* < 0.001 compared to control group.

**Table 3 ijerph-17-08887-t003:** Median score values MET-hours/week (MET-h/week) for the self-administered Pregnancy Physical Activity Questionnaires (PPAQs) during the first, second and third trimesters by activity intensity and type.

	First Trimester	Second Trimester	Third Trimester	Trimester Effect
Total MET-h/week	274.3 (177.9–400.5)	191.4 (131.3–259.17)	115.5 (92.4–169.2)	*
By intensity				
Sedentary	66.6 (43.5–91.7)	49.7 (35.0–76.2)	43.5 (18.8–69.1)	*
Light	112.8 (74.6–162.7)	79.8 (39.6–117.6)	52.6 (23.1–72.6)	*
Moderate	76.7 (33.2–146.0)	48.5 (18.6–84.6)	19.2 (7.0–39.3)	*
Vigorous	1.6 (0.0–4.8)	0.0 (0.0–1.6)	0.0 (0.0–0.0)	*
By type				
Household/caregiving	116.9 (72.4–171.5)	79.1 (38.8–122.4)	48.3 (23.0–73.4)	*
Occupational	62.1 (0.0–111.1)	0.0 (0.0–63.5)	0.0 (0.0–0.0)	*
Sports/Exercise	14.1 (5.9–29.8)	8.8 (2.5–21.5)	6.1 (0.8–13.1)	*

Data on PPAQ: *n* = 75; * *p* < 0.001 Trimester effect.

**Table 4 ijerph-17-08887-t004:** Median score values MET-hours/week (MET-h/week) for the self-administered Pregnancy Physical Activity Questionnaires (PPAQs) in the two groups, during the first, second and third trimesters by activity intensity and type.

	Intervention			Control		
	First Trimester	Second Trimester	Third Trimester	First Trimester	Second Trimester	Third Trimester
Total MET-h/week	275.9 (198.3–421.6)	170.6 (130.2–254.4)	128.3 (96.3–183.7)	274.3 (160.4–390.9)	196.7 (144.4–259.7)	98.9 (78.5–152.2)
By intensity						
Sedentary	70.0 (48.0–92.2)	49.7 (35.7–76.2)	64.7 (36.4–78.7) *	56.1 (37.8–88.1)	50.7 (30.3–75.8)	22.7 (9.4–49.8)
Light	104.3 (66.2–170.2)	73.2 (30.1–125.0)	52.1 (20.4–73.2)	122.3 (80.5–156.3)	88.2 (62.1–110.9)	52.6 (27.6–70.0)
Moderate	89.3 (30.4–168.4)	45.7 (18.5–86.0)	18.5 (3.4–33.2)	66.3 (35.4–128.2)	53.7 (19.3–84.6)	20.0 (10.4–43.5)
Vigorous	1.6 (0.0–4.8)	0 (0–1.6)	0 (0–0)	1.6 (0–5.2)	0 (0–1.6)	0 (0–0)
By type						
Household/caregiving	126.4 (74.4–180.6)	74.6 (28.7–129.5)	36.1 (18.5–71.2)	94.8 (67.6–160.1)	87.6 (48.1–120.2)	53.5 (29.6–74.7)
Occupational	65.4 (6.2–112.0)	0 (0–68.4)	0 (0–0)	35.8 (0–111.1)	0 (0–57.8)	0 (0–0)
Sports/Exercise	14.2 (5.6–24.7)	8.5 (3.4–13.8)	7.9 (0.4–13.2)	13.3 (6.4–31.4)	8.8 (1.7–22.5)	6.0 (0.8–12.9)

Data on PPAQ: *n* = 36 and 39 for intervention and control groups, respectively; * *p* < 0.001 Significant difference/control group.

**Table 5 ijerph-17-08887-t005:** Number of steps/hour (*n* = 18 and 14 for the intervention and control groups, respectively).

	Intervention			Control			
	First Trimester	Second Trimester	Third Trimester	First Trimester	Second Trimester	Third Trimester	*p*
Number of steps/hours	477 ± 61	408 ± 53	335 ± 31	575 ± 69	483 ± 60	378 ± 35	>0.5

**Table 6 ijerph-17-08887-t006:** Maternal outcomes in the intervention and control groups.

	Intervention	Control	OR (95%CI) Intervention/Control	*p*
Excessive GWG *n* (%)	16 (40.0)	17 (39.5)	1.02 (0.42–2.45)	0.5
GDM *n* (%)	8 (21.6)	4 (10.3)	2.41 (0.66–8.83)	0.1
Fasting plasma glucose (g/l)			Group: 0.04; Time: 0.6 Group*time: 0.3	
Trimester 1	0.77 ± 0.05	0.76 ± 0.06		
Trimester 2	0.80 ± 0.08	0.76 ± 0.06		
Trimester 3	0.79 ± 0.09	0.76 ± 0.06		
OGTT (g/l)			Group: 0.5; Time: <0.001 Group*time: 0.3	
T0	0.80 ± 0.08 1.23 ± 0.29	0.76 ± 0.06 1.25 ± 0.23		
T60				
T120	1.15 ± 0.28	1.10 ± 0.20		
Area under the curve (AUC)	204 ± 42	201 ± 32		0.7
Maternal hypertension during labor, *n*/%	7 (16.3)	9 (21.4)	0.71 (0.23–2.13)	0.3
Cesarean	7 (15.9)	6 (12.5)	0.75 (0.23–2.44)	0.4

Glycemia data are reported in mean and standard deviation (SD). Data on excessive gestational weight gain (GWG): *n* = 40 and 43 for the intervention and control groups, respectively. Data on gestational diabetes, glycemia values: *n* = 37 and 39 for the intervention and control groups, respectively. Cesarean data: *n* = 44 and 48 for the intervention and control groups, respectively. Maternal hypertension data: *n* = 43 and 42 for the intervention and control groups, respectively. (Odds Ratio (OR); Confidence Interval (CI).  * Group-by-time = interaction.

**Table 7 ijerph-17-08887-t007:** Neonatal outcomes in intervention and control groups.

	Intervention	Control	OR (95%CI) Intervention/Control	*p*
Gestational age at delivery (WA)	39.4 (39.4–41.1)	39.4 (39.4–39.4)		0.7
Preterm birth, *n* (%)	2 (4.5)	3 (6.1)	1.37 (0.21–8.60)	0.5
Weight, kilograms			Group: 0.1; Age effect: <0.001 Group *age effect: 0.9	
Birth	3.09 ± 0.41	3.23 ± 0.47		
First month	4.08 ± 0.54	4.27 ± 0.64		
Second month	5.25 ± 0.74	5.43 ± 0.73		
Length, meter			Group: 0.1; Age effect: <0.001 Group *age effect: 0.1	
Birth	49.6 ± 2.24	50.0 ± 1.96		
First month	53.0 ± 2.54	53.8 ± 3.24		
Second month	56.8 ± 2.27	58.3 ± 3.62		

WA: Weeks of amenorrhea, Delivery data: *n* = 44 and 49 for the intervention and control group, respectively. Data on weight and height: *n* = 33 and 27 for intervention and control groups, respectively. (Odds Ratio (OR); Confidence Interval (CI).  * Group-age = interaction.

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
