# Peer review of "Prenatal Counseling throughout Pregnancy: Effects on Physical Activity Level, Perceived Barriers, and Perinatal Health Outcomes: A Quasi-Experimental Study"

_ijerph, 2020, doi:10.3390/ijerph17238887_

Round 1
Reviewer 1 Report
Abstract: Second sentence seems incorrect (lines 14,15). " women report perceived barriers and overall the physical activity level is insufficient". Please correct so it is easily understood.
Background:
line 37 - modifiable 'behavior'?
Mention more about reference 7 - guadeloupe study - what was the population size? was it generalizable enough to consider as a rationale for this study?
line 46 - 'increasing or maintaining' seems incorrect. it can be either increasing low PA levels, or maintain high PA levels. please correct his.
Methods:
line 53 - Mention in 2-3 sentences about the PA promotion project.
line 57 - " The health professionals randomized into the intervention group". please clarify - randomized what? themselves? or do you mean 'were randomized'?
Was index pregnancy or parity considered in the study?
line 122 - Mention more detail about the questionnaire. Was it validated or standardized? mention the validity.
Author Response
Response to Reviewer 1 Comments
Dear reviewer,
We are grateful for the time you spent reviewing our manuscript and for sharing your expertise in providing your feedback. Thank you very much. We read the comments and appreciations carefully. It is our pleasure to submit to your proficiency our answers and propositions for the revised version of this work. All modifications in the manuscript have been specified in blue color.
Abstract:
Second sentence seems incorrect (lines 14,15). " women report perceived barriers and overall the physical activity level is insufficient". Please correct so it is easily understood.
Response 1: We have taken this remark into account in the abstract. A modification of the sentence has been made in the manuscript (lines 14-16), and the sentence should be easily understood.
Background:
line 37 - modifiable 'behavior'?
Response : We thank the reviewer for their question/comment because it is correct. It is indeed physical activity as modifiable behavior. We have added the word "behavior" in the sentence (lines 38,39).
Mention more about reference 7 - Guadeloupe study - what was the population size? was it generalizable enough to consider as a rationale for this study?
Response : Regarding the study by Atallah et al., [1], the population size was 685 people aged 15 or over, and is not representative of the population of Guadeloupe.
We agree that this may weaken the rationale for this study, however, there are few studies on physical activity behavior on this population. To our knowledge, it is the only study. We modified the presentation of the argument linked comment by the reviewer (lines 45, 46). A recent study reported on the rate of high overweight in this population, with a larger sample [2] compared the previous study by Atallah et al. We have added it in our manuscript.
line 46 - 'increasing or maintaining' seems incorrect. it can be either increasing low PA levels or maintain high PA levels. please correct his.
Response : We appreciate the suggestion from the reviewer. We have made a change in the manuscript related to this suggestion (line 49).
Methods:
line 53 - Mention in 2-3 sentences about the PA promotion project.
Response : We added some elements on PA promotion project which based on several elements reported in the literature such as low PA level during pregnancy, the perceived barriers, the lack of knowledge by health professionals [3–6]. These elements are reported in lines 56-62.
line 57 - " The health professionals randomized into the intervention group". please clarify - randomized what? themselves? or do you mean 'were randomized'?
Response : We agree that one sentence in our manuscript was not clear enough, especially on randomization. This point has been improved, and the interpretation should be easier to follow. Indeed, before beginning the research project, health professionals were randomly assigned to the control group or the intervention group. We did not want to influence the behaviors of the health professionals in the control group, who were free to either provide PA counseling to their patients or not. The intervention group attended two meetings, each lasting approximately 30 minutes, where they were informed of PA recommendations.
The pregnant women were allocated to either the intervention or control group, depending on the health professional who was following them.
Thank you for pointing out this weakness in our initial version. These elements have been corrected in the manuscript (lines 64- 67).
Was index pregnancy or parity considered in the study?
Response : In this study, parity was considered and we thank the reviewer for this relevant question. We chose not to present the parity in our table 1 "Baseline characteristics of study participants" because there was no significant difference between the two groups and there was no link between parity and the results.
line 122 - Mention more detail about the questionnaire. Was it validated or standardized? mention the validity.
Response : In response to the author's comment, we have provided more detail about the questionnaire created for "perceived barriers to physical activity". To our knowledge, there is no validated questionnaire in the literature. We used a structured questionnaire and based on the elements of the literature, such as other authors [7]. We put forward arguments on the structured questionnaire with references in the corrected version of the manuscript (lines 133-147).
We agree that this method could weaken the results in our study.
REFERENCES:
- Atallah A, Pitot S, Savin J, Moussinga N, Laure P. Physical activity carried out in a general population of Guadeloupe (FWI), determining factors: Results from APHYGUAD study. Science & Sports. 2012;27:160–8. doi:10.1016/j.scispo.2011.05.004.
- Carrère P, Fagour C, Sportouch D, Gane-Troplent F, Hélène-Pelage J, Lang T, et al. Diabetes mellitus and obesity in the French Caribbean: A special vulnerability for women? Women Health. 2018;58:145–59.
- Currie S, Sinclair M, Liddle DS, Nevill A, Murphy MH. Application of objective physical activity measurement in an antenatal physical activity consultation intervention: a randomised controlled trial. BMC Public Health. 2015;15:1259.
- Coll CVN, Domingues MR, Gonçalves H, Bertoldi AD. Perceived barriers to leisure-time physical activity during pregnancy: A literature review of quantitative and qualitative evidence. J Sci Med Sport. 2017;20:17–25.
- De Vivo M, Mills H. “They turn to you first for everything”: insights into midwives’ perspectives of providing physical activity advice and guidance to pregnant women. BMC Pregnancy Childbirth. 2019;19:462.
- McLellan JM, O’Carroll RE, Cheyne H, Dombrowski SU. Investigating midwives’ barriers and facilitators to multiple health promotion practice behaviours: a qualitative study using the theoretical domains framework. Implement Sci. 2019;14:64.
- Haakstad LAH, Vistad I, Sagedal LR, Lohne-Seiler H, Torstveit MK. How does a lifestyle intervention during pregnancy influence perceived barriers to leisure-time physical activity? The Norwegian fit for delivery study, a randomized controlled trial. BMC Pregnancy Childbirth. 2018;18:127.

Reviewer 2 Report
General comments
The authors undertake an important study on the feasibility of a low-cost counseling intervention to improve prenatal physical activity practice and antenatal healthcare. The findings are worthwhile in informing contextual-specific strategies to promote the uptake of physical activity during pregnancy. Educational intervention is key in addressing the barriers relating to lack of information or lack of knowledge.
Abstract
- Line 18: Add the setting or country after the Guadeloupe Hospital.
Introduction
- The found the introduction not capturing the context of the matter under investigation─Prenatal physical activity counseling and effect on physical activity level, perceived barriers, and perinatal health outcomes. I suggest this connection to be presented so as to orientate the readers sufficiently to the problem of the study
Methodology
- Line 76: I do not understand this “ (Visit 0).”
- Line 161: A repetition of Lines 152-153.
Results
- Line 216: Write “1 and 3” in words to maintain consistency.
- Line 239: Remove the “all” in the bracket
- Lines 266-299: Some of these findings could be summarise to 3-4 paragraphs.
- Line 282: Write “1” in words
- Line 298: I suggest you change the word “softened” to ‘reduce’
- Line 329: I suggest you create a separate heading here as ‘Limitations of the study’
Author Response
Response to Reviewer 2 Comments
Dear reviewer,
We are grateful for the time you spent reviewing our manuscript and for sharing your expertise in providing your feedback. Thank you very much. We read the comments and appreciations carefully. It is our pleasure to submit to your proficiency our answers and propositions for the revised version of this work. All modifications in the manuscript have been specified in blue color.
Abstract:
- Line 18: Add the setting or country after the Guadeloupe Hospital.
Response : Guadeloupe is a French department. At the request of the author, we have specified this in line 18.
Introduction
The found the introduction not capturing the context of the matter under investigation─Prenatal physical activity counseling and effect on physical activity level, perceived barriers, and perinatal health outcomes. I suggest this connection to be presented so as to orientate the readers sufficiently to the problem of the study
Response : We thank the reviewer for this remark. We have made changes to the introduction of the manuscript in the corrected version in blue color. We have moved the sentences to make reading easier and orientate readers.
Methods:
- Line 76: I do not understand this “ (Visit 0).”
Response : The word "visit 0" was defined as the first meeting between the healthcare professional and the patient where the coordinator was invited to explain what the research entailed. We have removed this word because it may be a misunderstanding for the reader (line 77).
2. Line 161: A repetition of Lines 152-153.
Response : We thank the reviewer for this comment. The repetition has been deleted for easy understanding.
Results
Line 216: Write “1 and 3” in words to maintain consistency.
Response : We thank the reviewer for these comments/observations. Corrections have been made (lines 221, 222).
Line 239: Remove the “all” in the bracket
Response : At the request of the reviewer, we have deleted "all" in the bracket (line 250).
Lines 266-299: Some of these findings could be summarise to 3-4 paragraphs.
Response : We have made changes at the request of the reviewer and have reduced the paragraphs (lines 278-310). Indeed, there is a better understanding of these findings for the readers after these modifications. We thank the reviewer for this comment.
Line 282: Write “1” in words
Response : At the request of the reviewer, a correction was made to line 293.
Line 298: I suggest you change the word “softened” to ‘reduce’
Response : At the request of the reviewer, a correction was made to line 309, and the word "softened" was changed.
Line 329: I suggest you create a separate heading here as ‘Limitations of the study’
Response : We have taken the author's remark into consideration. Considering that we also present strengths, we have added in heading "Limitations and strengths of the study" to line 347.
